# A Two-Branch CNN Fusing Temporal and Frequency Features for Motor Imagery EEG Decoding

**DOI:** 10.3390/e24030376

**Published:** 2022-03-08

**Authors:** Jun Yang, Siheng Gao, Tao Shen

**Affiliations:** School of Information Engineering and Automation, Kunming University of Science and Technology, Kunming 650504, China; yang-jun@kust.edu.cn (J.Y.); gaosiheng@stu.kust.edu.cn (S.G.)

**Keywords:** electroencephalography (EEG), motor imagery (MI), temporal and frequency features, convolutional neural network (CNN)

## Abstract

With the development of technology and the rise of the meta-universe concept, the brain-computer interface (BCI) has become a hotspot in the research field, and the BCI based on motor imagery (MI) EEG has been widely concerned. However, in the process of MI-EEG decoding, the performance of the decoding model needs to be improved. At present, most MI-EEG decoding methods based on deep learning cannot make full use of the temporal and frequency features of EEG data, which leads to a low accuracy of MI-EEG decoding. To address this issue, this paper proposes a two-branch convolutional neural network (TBTF-CNN) that can simultaneously learn the temporal and frequency features of EEG data. The structure of EEG data is reconstructed to simplify the spatio-temporal convolution process of CNN, and continuous wavelet transform is used to express the time-frequency features of EEG data. TBTF-CNN fuses the features learned from the two branches and then inputs them into the classifier to decode the MI-EEG. The experimental results on the BCI competition IV 2b dataset show that the proposed model achieves an average classification accuracy of 81.3% and a kappa value of 0.63. Compared with other methods, TBTF-CNN achieves a better performance in MI-EEG decoding. The proposed method can make full use of the temporal and frequency features of EEG data and can improve the decoding accuracy of MI-EEG.

## 1. Introduction

The brain-computer interface (BCI) constructs an information interaction channel between the human brain and the external environment, which is independent of normal neural pathways and muscles. It can directly read and analyze the bioelectrical signals generated by the brain and then convert the information into instructions to control external devices [1,2]. In the biomedical field, the neuroimaging techniques for recording brain neural activities mainly include depth electrodes, electrocorticography (EcoG), functional magnetic resonance imaging (fMRI), magnetoencephalography (MEG), and electroencephalography (EEG) [3]. Compared with other neuroimaging techniques, EEG is easy to obtain, and it has a low acquisition cost and high temporal resolution. As a result of its advantages, EEG is favored in the application of BCI technology. According to the acquisition methods of EEG signals, BCI can be divided into three types, namely invasive, partially invasive, and non-invasive BCI [4]. Although invasive BCI can obtain high-quality EEG signals, it requires the surgical placement of invasive devices in the brain. Therefore, it has certain risks and safety issues and is often unacceptable to experimental subjects. Non-invasive BCI can collect brain neural activities with the assistance of an EEG cap. It is simple to operate and has a low risk. Compared with invasive BCI, non-invasive BCI has obvious advantages [5,6].

The motor imagery (MI) paradigm is a psychological process that simulates motor intention without producing real motor behaviors, and it activates neural potential in the primary sensory motor area of the brain. Compared with evoked EEG, MI can spontaneously induce the potential activity in different brain regions without external stimulation, and it has high application value in the field of neural rehabilitation. A plethora of studies have found that μ rhythm (8–13 Hz) and β rhythm (17–30 Hz) are closely related to human motion perception in normal EEG signal bands. When subjects perform motor imagery activities, such as imagining limb movement on one side, the μ rhythm and β rhythm energy of the contralateral EEG rhythm decreases. When the subjects finish their motor imagery or are in a resting state, the μ and β rhythm energy increases. This phenomenon of energy change in a specific frequency band is referred to as event-related desynchronization (ERD) and event-related synchronization (ERS) [7,8,9]. Based on this phenomenon, BCI can be used as a medical rehabilitation approach for disabled users [10], and it has a positive impact on the quality of life and external environment interaction of normal users. Besides, BCI also provides new human–computer interaction modes, such as computer game control, music generation and virtual reality. More and more BCIs can control complex devices [11,12], including artificial limbs, mobile robots and mechanical arms.

Although the BCIs based on the motor imagery paradigm develop rapidly, there are many limitations in their classification and recognition. Due to the defect of the non-invasive acquisition method and the uncertainty of EEG traceability, EEG signals have a low signal noise ratio (SNR) and low spatial resolution [13,14,15]. Meanwhile, EEG signals show non-stationarity over time and physiological differences between different individuals, which requires a time-consuming calibration process before BCIs can be used [16]. In view of these limitations, different feature extraction methods and classification methods have been studied for motor imagery task recognition. Using the spatial correlation of EEG signals, spatial filtering can eliminate the noise of EEG signals and locate local cortical neural activities. Common spatial pattern (CSP) is a widely used spatial feature extraction method in motor imagery tasks [17,18], and it achieves good performance in much of the research on motor imagery task recognition [19,20,21]. Researchers have proposed many improved methods based on CSP to improve the accuracy of EEG decoding. Kim et al. used CSP and linear discriminant analysis (LDA) to classify MI tasks [22]. Kai et al. proposed the filter bank common spatial pattern (FBSCSP), which segments EEG signals with multiple band-pass filters and spatial filters to calculate mutual information of CSP features in different sub-frequency bands [23]. Thang et al. used four different machine learning methods, including LDA, QDA, LSVM and RBFSVM, to verify the decoding performance of MI-EEG [24]. Zhang et al. proposed a temporally constrained sparse group spatial pattern (TSGSP), which optimizes the filtering frequency band and time window in CSP [25].

In recent years, the EEG classification methods based on deep learning (DL) have been widely used [26,27]. The convolution neural network (CNN) is one of the deep learning models. In the field of computer vision, the lower-level convolution layers learn the general feature representations of data, whereas the higher-level ones learn the specific feature representations. Then, the parameters and weights of each layer are updated by the backpropagation algorithm. Attributed to its end-to-end model structure and automatic feature extraction capability, CNN can avoid the interference of redundant information and improve the classification performance. In this way, the corresponding features can be effectively learned from the data, and the complicated feature extraction process is avoided. CNN has been applied to analyze EEG data. Yang et al. designed an end-to-end CNN framework that uses one-dimensional convolution kernels with sizes of 1 × m and n × 1 to learn the temporal and spatial features from the original MI-EEG data [28]. Tabar et al. combined CNN with a stacked automatic encoder (SAE) to construct a new depth network for EEG decoding. In this method, the convolution layer is used to extract the time-frequency features obtained from short-time Fourier transform (STFT), and the convolution kernel and network parameters are trained [29]. Zhang R et al. proposed a hybrid deep learning framework that combines CNN and LSTM to simultaneously learn the spatial and temporal characteristics of MI-EEG samples [30]. Hou et al. combined the EEG source imaging (ESI) method with the continuous wavelet transform to obtain two-dimensional features with a high time-frequency resolution from the traceability signal from the scout. Then, the obtained features are decoded by CNN [3]. Chen et al. proposed a filter bank spatial filtering and spatio-temporal convolution neural network (FBSF-TSCNN) for MI-EEG decoding, in which TSCNN blocks can learn spatio-temporal features from the original MI-EEG [31]. Dai et al. proposed a hybrid convolution scale CNN model for MI-EEG decoding that uses convolution kernels with different sizes to extract the temporal and spatial features of EEG signals [32].

At present, most CNN-based MI-EEG decoding methods use a single input, such as the original EEG signal, CSP feature and time-frequency graph. However, the CNN with a single-branch input cannot make full use of the effective EEG information, which reduces the performance of the decoding model. To address this issue, this paper proposes a two-branch CNN model called TBTF-CNN, which takes EEG signals and time-frequency graphs as input to utilize the temporal and frequency features of EEG. First, the original EEG signal has high-resolution temporal information, and the identification features can be extracted by spatio-temporal convolution. Then, the features extracted from EEG signals through continuous wavelet transform (CWT) include time, frequency and spatial information. Finally, CNN fuses the features of the two branches and inputs them into the classifier. When updating the trainable parameters of the CNN network, only the features related to MI tasks are concerned, and the classification accuracy is improved by combining the features of EEG data with time-frequency data. The experimental results on the BCI competition IV 2b dataset show that the proposed TBTF-CNN model achieves a good performance, which proves the effectiveness of the architecture.

## 2. Methods

### 2.1. System Framework

The system framework is shown in Figure 1. In MI-EEG data, μ rhythm (8–13 Hz) and β rhythm (17–30 Hz) show different energy changes in related electrodes under different motor imagery states, with emphasis on C3, Cz and C4 electrodes. In the data preprocessing stage, the MI-EEG signals of C3, Cz and C4 channels are selected, and the raw data are filtered by a fifth-order Butterworth FIR bandpass filter to obtain the motor imagery EEG signals of 8–13 Hz and 17–30 Hz. The time-frequency feature of each channel is obtained by CWT. Then, the EEG signals of μ rhythm and β rhythm are spliced as input features of a CNN branch. The feature expression form is N × T, where N represents the number of channels, and T represents the number of sampling points. In the same way, the time-frequency features of the two frequency bands are spliced according to the corresponding relationship, and the feature expression form is a two-dimensional time-frequency diagram. Subsequently, the time-frequency diagram is input into another CNN branch as Appendix A to the original EEG. The two branches of CNN extract features from the input EEG signal and time-frequency graph, respectively. After the process of convolution and pooling, the features output by the two branches are flattened and input into the fusion layer. Finally, the fusion layer combines the features of the two branches into a one-dimensional vector and inputs it into the classifier to obtain the prediction results.

### 2.2. MI-EEG Feature Representation

The EEG decoding based on MI-EEG consists of three steps: data preprocessing, feature extraction and classification. Among them, suitable feature extraction and reasonable identification feature expression can improve the accuracy and efficiency of EEG decoding, which is the premise of EEG analysis. The original MI-EEG data are usually in the form of a two-dimensional matrix consisting of signal channels and sampling points, and each row of the matrix represents the sampling data of each channel. This form of MI-EEG data contains spatial and temporal features. When the subjects perform motor imagery tasks, ERD/ERS phenomena can be observed at C3, Cz and C4 electrodes. Thus, the EEG signals of the three channels are selected for left-hand and right-hand MI-EEG decoding in this study. As a result of the different decoding methods, most studies directly filter the original signal with a frequency of 8–30 Hz and use the EEG in this frequency band for subsequent decoding. Inspired by the filter bank common spatial pattern (FBCSP) feature extraction method, this study divides the original signal into frequency bands of 8–13 Hz and 17–30 Hz to learn the identification features from the decoding model. As shown in Figure 2, the EEG data of the two sub-bands are combined into a 6 × T 2D matrix sequentially. The first three channels are MI signals with μ rhythm in C3, Cz and C4, and the last three channels correspond to the signals with β rhythm.

In this study, CWT was employed to transform the EEG data in each channel into a two-dimensional time-frequency energy map. As shown in Figure 3, the time-frequency diagrams of each channel were combined as the Appendix A of the EEG data. Compared with wavelet transform, Fourier transform can only obtain the frequency domain features of signals, while STFT loses part of the timing features due to its fixed window length. Wavelet transform is a time-frequency localization analysis that can focus on any details of the signal by shifting and stretching operations to gradually refine the signal at multiple scales. This study chose the “Morlet” wavelet for continuous wavelet transform, and the EEG signal was processed through Equations (1)–(3) to obtain a two-dimensional time-frequency energy map.
(1)$$\begin{array}{c} \Phi \left(t\right)=e^{i\omega t}*e^{-\frac{t^{2}}{2}}, \end{array}$$
(2)$$\begin{array}{c} \Phi_{\alpha ,\beta }\left(t\right)=\frac{1}{\sqrt{\alpha }}e^{i\omega \frac{\left(t-\beta \right)}{\alpha }}*e^{-\frac{{\left(t-\beta \right)}^{2}}{2\alpha^{2}}}, \end{array}$$
(3)$$\begin{array}{c} CWT\left(\alpha ,\beta \right)=\frac{1}{\sqrt{\alpha }}\int_{-\infty }^{+\infty }f\left(t\right)\Phi_{\alpha ,\beta }\left(t\right)dt. \end{array}$$
where ω represents the wavelet center frequency; t represents time; α is the scale transformation factor; and β is the time translation factor.

### 2.3. Proposed TBTF-CNN Architecture

CNN is a feedforward neural network that involves convolution-pooling computation. Its network structure consists of two parts, i.e., the feature extraction layer formed by the combination of convolution layers and pooling layers, and the classification output layer composed of several fully connected layers [33]. In the convolution part, the input 3D tensor was convolved with the convolution kernel, and then an activation function $f\;\left(a\right)$ was used to output the feature map. The feature map of each convolution layer can be expressed as:(4)$$\begin{array}{c} h_{ij}^{k}=f\left(a\right)=f\left({\left(W^{k}*x\right)}_{ij}+b_{k}\right). \end{array}$$
where x represents the input data; $W^{k}$ represents the weight matrix of the k-th convolution kernel; $b_{k}$ corresponds to the bias of convolution kernel k; i and j represent the number of adjacent convolution layers. The fully connected layer is composed of several independent neurons, and the neurons between the layers are fully connected. The results of data passing through the fully connected layer can be expressed as:(5)$$\begin{array}{c} y=f\left(W_{y}x+b_{y}\right). \end{array}$$
where $W_{y}$ denotes the weight matrix of the layer, $b_{y}$ denotes the bias and $f\left(\cdot \right)$ denotes the activation function.

Figure 4 shows the structure of the proposed TBTF-CNN. At the input end, the EEG data and time-frequency diagram data were reconstructed into a three-dimensional tensor (height × width × channel).

The EEG branch extracts spatial and temporal features from the data of α band and β band. The branch consists of a convolution layer, a max pooling layer and a flatten layer. EEG is a king of time series data, and its structure is usually electrode × sampling point. Some studies use CNN to decode EEG [34,35] data with the input form of electrode × sampling point × 1. These studies used horizontal and vertical 1D convolution kernels to divide convolution operations into temporal convolution and spatial convolution to improve the decoding performance. After temporal convolution and spatial convolution, the feature map is in the form of 1 × Nw × Nf, where Nf represents the number of convolution kernels and 1 × Nw represents the one-dimensional horizontal vector feature map. In this study, the computation of temporal convolution and spatial convolution was simplified. Meanwhile, the input data of EEG were reconstructed into 1 × T × 6, where 1 corresponds to the height of the convolution layer; T represents the number of sampling points, and it corresponds to the width of the convolution layer; 6 represents the total number of C3, Cz and C4 electrodes in two sub-bands, and it corresponds to the number of channels in the convolution layer. A one-dimensional convolution kernel along the horizontal axis was used to extract the features of each channel. The size of the convolution kernel was set to 1 × 8, and the step size was 1 × 8. The output EEG data passing through this layer had a dimension of 1 × Nw × Nf, which is the same as the dimension of the feature map extracted by temporal convolution and spatial convolution. The experimental results show that the results obtained by this simplified method are basically the same as those obtained by the original method. Besides, the exponential linear units (ELUs) function [36] was used as the activation function of the convolution layer, which can accelerate the learning speed and improve the accuracy of the neural network. It can be expressed as follows:(6)$$\begin{array}{c} f\left(x\right)=\left\{\begin{array}{c} x,\;\\ a\left(e^{x}-1\right),\; \end{array}\begin{array}{c} \;if\;x\geq 0\\ \;if\;x<0 \end{array}\;a\right.>0 \end{array}$$Then, a max-pooling layer was used to down-sample the data of the convolution layer. The size of the pooling kernel is 1 × 3, and the step size is 1 × 3. Finally, the flatten layer converted the features of the EEG branch into a one-dimensional vector.

The CWT branch extracts the time-frequency features from the data of α and β bands. Its network structure is the same as that of the EEG branch, which consists of a convolution layer, a max-pooling layer and a flatten layer. We reconstructed the time-frequency diagram into the form of 64 × 93 × 1, which represents the gray-scale image tensor with the size of 64 × 93. Different from the two-dimensional convolution kernel commonly used in other studies, we used a one-dimensional convolution kernel along the vertical axis to extract time-frequency features, which is similar to Tarbar et al.’s [29] time-frequency diagram using short-time Fourier transform to extract features, because the time-frequency diagram contains time, frequency and electrode position information. The one-dimensional vertical convolution kernel with the same height as the input data can better extract time-frequency features by sliding horizontally along the time axis. The size of the convolution kernel is 64 × 1 and the step size is 1 × 1. Besides, the ReLU function is used as the activation function of the convolution layer, and it is expressed as follows:(7)$$f\left(x\right)=\left\{\begin{array}{c} x,\;\\ 0, \end{array}\begin{array}{c} \;if\;x\geq 0\\ \;if\;x<0 \end{array}\right..$$The size of the pooling kernel is 1 × 3, and the step size is 1 × 3. The time-frequency features output by the pooling layer are converted into one-dimensional vectors by the flatten layer.

In this study, the features extracted from the two branches were spliced into a one-dimensional feature vector, which was then used as the input of the fully connected layer. The classification part consists of two fully connected layers. The first layer contains 128 neurons, and the activation mode is ReLU. There are two neurons in the output layer. The softmax function maps the output y of the previous layer to the prediction probability p, and the output is:(8)$$p\left(l_{i}|y\right)=\frac{e^{y}}{\sum_{i=1}^{N}e^{y}}.$$

During the input phase, the EEG data and time-frequency map data were normalized to enhance the data concentration and eliminate the adverse effects of sample outliers. Meanwhile, a “dropout” layer was added between the fully connected layer to effectively suppress the overfitting problem. In the training process of the model, the Adam optimizer [37] with $\beta_{1}$ = 0.9 and $\beta_{2}$ = 0.999 was used to update the trainable parameters of each network layer, and the initial learning rate was set to 0.001. See Table 1 for detailed network parameters.

## 3. Experimental Results

### 3.1. Experimental Dataset

In the experiment, the BCI competition IV 2b dataset is used to evaluate the decoding performance of the proposed model. This dataset is provided by the Graz University of Technology [38], and it is very famous in the field of motor imagery EEG datasets. BCI competition IV 2b datasets contains the EEG data of nine healthy subjects, and the recording process involves no moral and ethical issues. For each subject, five sessions of motor imagery EEG signals were recorded. The first two sessions are routine cue motor imagery, and the last three sessions include feedback. In each session, three electrodes, i.e., C3, Cz and C4, were used to collect the motor imagery EEG of the left hand and right hand, and the sampling frequency was 250 Hz. As shown in Figure 5, each experiment begins with a fixation cross “+” and an acoustic warning tone. When t = 3 s, the arrow pointing to the left or right appeared on the screen for 1.25 s. Then, the subjects were asked to imagine the movement of the corresponding hand within 4 s, and the experiment ended with a rest of 1.5 s. After EEG collection, the influence of artifacts on each trial was evaluated by experts, and the trials with artifacts were recorded as “1023” events. In the annotation of the dataset, “0” corresponds to clean experiments and “1” corresponds to the experiments with artifacts. The experimental paradigm of the last three sessions with smile feedback cues is similar to that of routine cues, with 120 trials in each of the first two sessions and 160 trials in each of the last three sessions. In this study, the motor imagery EEG data of the first three sessions are selected for model training and evaluation.

### 3.2. Performance of the Proposed TBTF-CNN

In this study, the data at 0.5 s–4 s after the start of the motor imagery experiment are taken as the EEG data of each motor imagery task, so the data format of a single trial is 3 × 875. To make full use of all the experimental data, 10-fold cross-validation is performed to evaluate the classification performance of the model. The EEG data of a subject are divided into 10 equal subsets. Then, one subset is randomly selected as the test set, and the other nine subsets as the training set. The training is repeated 10 times, and the average of the training results is taken as the classification accuracy of a single subject. The experimental platform is based on the Tensorflow deep learning framework, and GPU GeForce 2080 is used to speed up the network training process.

To evaluate the performance of the two-branch input model, the experimental results of the proposed TBTF-CNN model are compared with those of the single-branch CNN model. The single-branch CNN model is set as EEG-CNN of the EEG branch and CWT-CNN of the CWT branch. In this case, the network has only one input form, which helps to verify the effectiveness of inputting EEG data and CWT data simultaneously in MI-EEG decoding. As shown in Figure 6a, the structure of the EEG branch is fine-tuned so that the network can adapt to the input of EEG data, and too many parameters will make the model unable to fit. Figure 6b shows the CWT branch, where the temporal feature learning branch is removed from the original model to evaluate the influence of time-frequency features on the decoding results.

The average classification accuracy of nine subjects is tested under different branch structures, and the experimental results are shown in Figure 7. It can be seen that the average classification accuracy of EEG-CNN is 70.56%, which is lower than that of other models. The results show that, in the shallow CNN structure, only learning the temporal features of EEG data is not conducive to the decoding of MI tasks. The average classification accuracy of CWT-CNN reaches 77.22%, which is not much different from the result of Tabar et al. [29] using STFT to extract features from the EEG data. For non-stationary EEG signals, CWT has a tunable time-frequency window, which can adjust according to the frequency. Therefore, CWT is more advantageous in extracting the time-frequency features of EEG signals. The experimental results also show that CWT-CNN has fewer parameters and improves the training and decoding efficiency. Compared with the single-branch CNN model, the proposed TBTF-CNN model achieves an average classification accuracy of 81.3% on nine subjects, which is higher than that of EEG-CNN and CWT-CNN. This shows that it is effective to input the EEG signal and its corresponding time-frequency map for EEG decoding in MI tasks. The CWT branch achieves a higher accuracy than the EEG branch, which proves that time-frequency feature expression plays an important role in EEG decoding. This study selects subjects 4, 5, 6 and 8 with high classification accuracy and visualizes their EEG time-frequency feature maps through the CWT branch to observe the time-frequency relationship in specific areas. As shown in Figure 8, the feature map obtained by the “Morlet” wavelet transform contains the joint distribution information in the time domain and frequency domain, which clearly describes the frequency and time relationship of the EEG signal. Meanwhile, it can be seen from the figure that only part of the time-frequency feature map after the convolutional layer has a high color brightness, indicating that different MI tasks are only sensitive to specific time and frequency.

According to the experimental results of the three models, the confusion matrix is shown in Figure 9 to evaluate the recall rates of different models. In the confusion matrix, the bottom right represents the average classification accuracy of nine subjects, and the other evaluation indexes correspond to the average recall rate and average accuracy of the two classes of tasks. It can be seen from Figure 9 that the TBTF-CNN model obtains a recall rate of 76.6% and 85.9% and a precision of 84.5% and 78.6% for the left-hand class and the right-hand class. The results show that the TBTF-CNN architecture performs better than the single-branch CNN model.

Then, for different models, the kappa coefficient [39] of each subject is calculated, and the result is shown in Figure 10. The kappa coefficient can eliminate the interference of random classification, and it is a common index to measure classification accuracy. Its calculation is as follows:(9)$$k=\frac{p_{0}-p_{e}}{1-p_{e}},$$
where $p_{0}$ represents the number of correctly classified samples in each class divided by the sum of all samples, i.e., the average classification accuracy of each subject and $p_{e}$ represents the random classification probability [40]. The $p_{e}\;$ value is set to 0.5 because this experiment is a two-class MI task. It can be seen from Figure 10 that the average kappa value of the EEG-CNN architecture is 0.41, and that of CWT-CNN is 0.54. The proposed TBTF-CNN achieves the largest kappa value of 0.63 among all the comparison experiments. The results show that TBTF-CNN has a better decoding performance than the single-branch CNN.

In addition, a machine learning comparison experiment is carried out to evaluate the performance of TBTF-CNN. CSP is a widely used feature extraction method in MI tasks. It can effectively construct the optimal spatial filter to distinguish EEG signals for two classification tasks in MI-EEG [18]. In this experiment, the machine learning model of CSP-LDA is taken for comparison, which achieves a good performance in previous MI-EEG decoding. To make a fair comparison, the performance is evaluated based on 10-fold cross- validation. The dataset is divided in the same way, and the expression of the EEG data is the same as that described in Section 2.2. The results of CSP-LDA are shown in Figure 7 and Figure 10. The average classification accuracy of CSP-LDA on nine subjects is 71.29%, and the kappa value is 0.426. The experimental results show that the decoding performance of CSP-LDA is 0.73% better than that of EEG-CNN and 5.93% worse than that of CWT-CNN. The proposed TBTF-CNN achieves the best performance, which is 9.71% higher than that of CSP-LDA. Figure 11 presents the MI-EEG feature distribution of nine subjects obtained by the CSP algorithm. The red dots and blue dots in the diagram represent the MI features of the left hand and right hand, respectively. Combined with the scatter plot, the features of subjects 1, 4, 5, 6, 8 and 9 have obvious discrimination, so a high average classification accuracy is achieved for these subjects. The features of subjects 2, 3 and 7 cannot be well separated, and this may be due to the distraction of these subjects during the experiment.

### 3.3. Comparison with Other Methods

In this section, the proposed model is compared with the models proposed in the literature to evaluate the overall performance of TBTF-CNN. Table 2 summarizes the average classification accuracy obtained by different methods for nine subjects of the dataset, where the highest accuracy of each subject is highlighted in bold font. It can be seen from the table that the proposed TBTF-CNN model achieves the best performance for most subjects and obtains the highest average classification accuracy. Specifically, the classification accuracy of Subject 1 and Subject 6 is increased by 14.5% and 4%, respectively. All of the comparison results show that our method achieves a better performance than other methods.

## 4. Discussion

According to the feature scatter diagram (shown in Figure 11), the subjects with an average classification accuracy higher than 80% have highly differentiated EEG features. Among them, Subject 4 and Subject 5 have the most obvious discrimination between the two classes of MI-EEG features, and the classification accuracy of these two objects is 98.1% and 89.7%, respectively. After CSP feature extraction, the features of Subject 2 and Subject 3 cannot be well separated, so the classification accuracy of these two subjects is only 63.3% and 62.3%. The reason for this phenomenon may be that the subjects are interfered with by external factors or their own factors, leading to the subjects’ inability to perform effective motor tasks. The classification performance in Figure 7 indicates that shallow CNN decodes the original EEG data poorly, and the average accuracy of MI tasks is 70.7%. CNN performs well in the image field, and CWT is suitable for non-stationary EEG signals. Therefore, the performance of the decoding model is improved by taking time-frequency graph data as an input, and the average classification accuracy is 77.4%. The proposed TBTF-CNN model uses both EEG data and time-frequency graph data as an input, so the network can further learn the temporal features and frequency features of the data, and the classification accuracy is improved. As for the machine learning model CSP-LDA, its classification accuracy is 71.5%, which is lower than that of most CNN architectures, indicating that deep learning has great advantages in decoding MI-EEG. The confusion matrix in Figure 9 shows that the recall rate of the three CNN models for right-handed imagination tasks is higher than that of left-handed imagination tasks. Among them, the recall rates of TBTF-CNN and CWT-CNN reach 85.9% and 84.3%, respectively. The proposed TBTF-CNN model performs more uniformly and better than other CNN architectures in all evaluation indexes, which proves the robustness of the model. The kappa coefficient shown in Figure 10 indicates that TBTF-CNN and CWT-CNN have higher kappa values, i.e., 0.63 and 0.55, respectively. This result further shows that the proposed TBTF-CNN model has a high robustness. Compared with other methods (as shown in Table 2), Ang et al. [21] used the FBCSP algorithm to divide EEG data into multiple frequency bands to extract features and obtain high classification accuracy. Tabar et al. [29] also divided the data into 6–13 Hz and 17–30 Hz for decoding. This shows that dividing EEG data into multiple sub-bands can improve the decoding performance of MI-EEG. Huang et al. [41] and Tabar et al. [29] both used STFT to extract the time-frequency features of EEG, and the classification accuracy was 73.9% and 77.6%, respectively. Since the time-window length of STFT is fixed, it cannot adjust to non-stationary EEG signals. Therefore, CNN-SAE is exploited to improve the learning ability of time-frequency features of the model, which increases the training parameters of the model and reduces the decoding efficiency. Although EEG features are expressed in time-frequency graphs, it can be seen from the results of this study that using CWT to extract time-frequency features achieves a higher average classification accuracy. Zheng et al. [42] and Raza et al. [43] both used the machine learning method to decode MI-EEG, and the classification accuracy was 78% and 69.7%, respectively. The machine learning method needs to conduct complex feature extraction calculation, whereas CNN has end-to-end structure characteristics, which simplifies the feature extraction and classification process of EEG data, and the classification performance is better.

EEG is a time-varying non-stationary signal, and a suitable time-frequency analysis method is very important to improve the performance of the decoding model. Due to the trade-off between the time and frequency resolution, it is difficult for STFT to generate time-frequency maps with high-quality information. This may degrade the classification performance of the BCI system. For MI-EEG signals lasting 3–5 s, CWT has a superior time-frequency resolution, and can express the signal more clearly without losing the information of time and electrode-frequency [44]. Compared with the STFT method, the CWT method achieves a better classification performance, thus showing the feasibility of MI-BCIs.

## 5. Conclusions

In this study, the TBTF-CNN model is proposed for MI-EEG decoding. TBTF-CNN has two input branches that can adequately learn the temporal and frequency features of EEG data and improve the classification accuracy. Meanwhile, the EEG data are reconstructed to simplify the process of spatio-temporal feature extraction, and continuous wavelet transform is used to express time-frequency features. The experimental results on the BCI competition IV 2b dataset show that TBTF-CNN achieves an average classification accuracy of 81.3%. The proposed method provides a new way to improve the decoding performance of BCI systems.

## 6. Future Work

With the development of hardware in the near future, we will apply the proposed TBTF-CNN model to online BCI systems based on MI-EEG to verify its robustness and performance. Moreover, we will use data augmentation techniques to improve the quality and quantity of EEG data to investigate the statistical differences in the accuracy of different models.

## Figures and Tables

**Figure 1 entropy-24-00376-f001:**
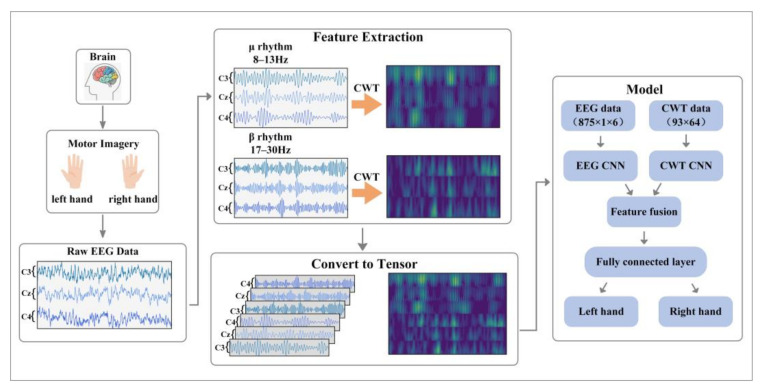
The system framework, including data preprocessing, feature extraction, feature representation and classification model.

**Figure 2 entropy-24-00376-f002:**
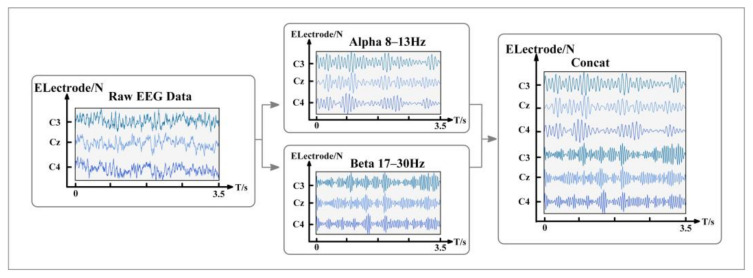
Temporal feature representation of two sub-bands EEG.

**Figure 3 entropy-24-00376-f003:**
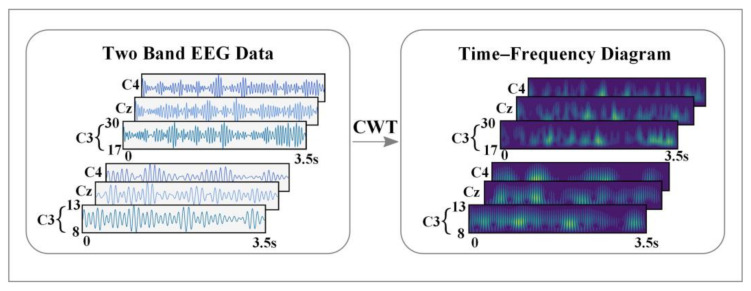
Time-frequency feature representation of two sub-bands EEG.

**Figure 4 entropy-24-00376-f004:**
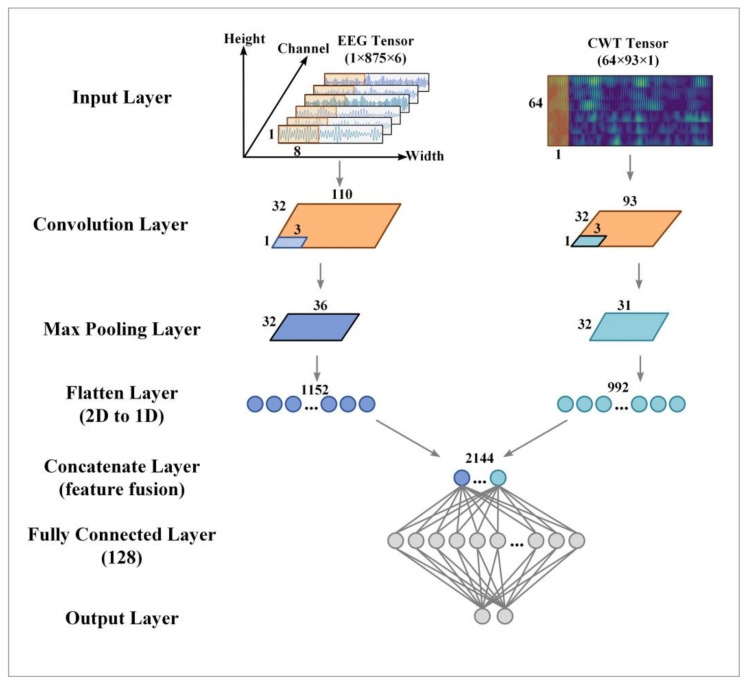
The structure of the proposed TBTF-CNN.

**Figure 5 entropy-24-00376-f005:**
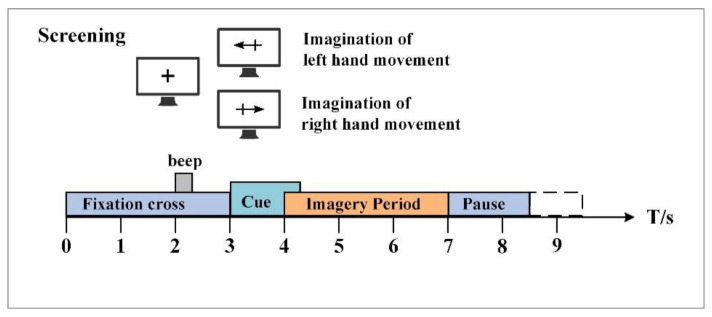
The experimental paradigm of the BCI competition IV 2b dataset.

**Figure 6 entropy-24-00376-f006:**
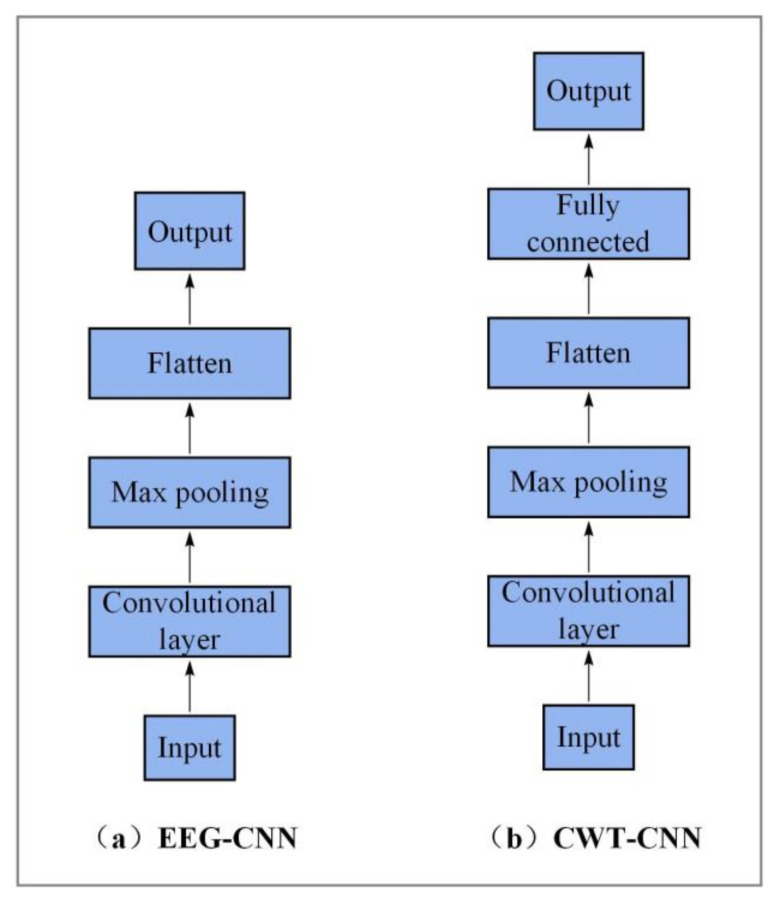
Network structure of the modified networks. (**a**) denotes an EEG-CNN branch to which only EEG data are input, and (**b**) denotes a CWT-CNN branch to which only CWT data are input.

**Figure 7 entropy-24-00376-f007:**
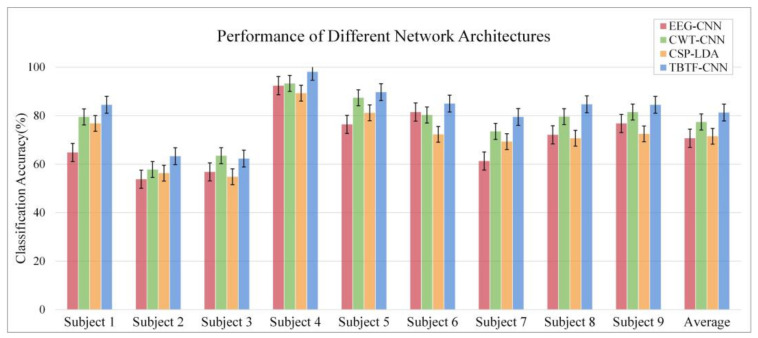
Average classification accuracy of each subject.

**Figure 8 entropy-24-00376-f008:**
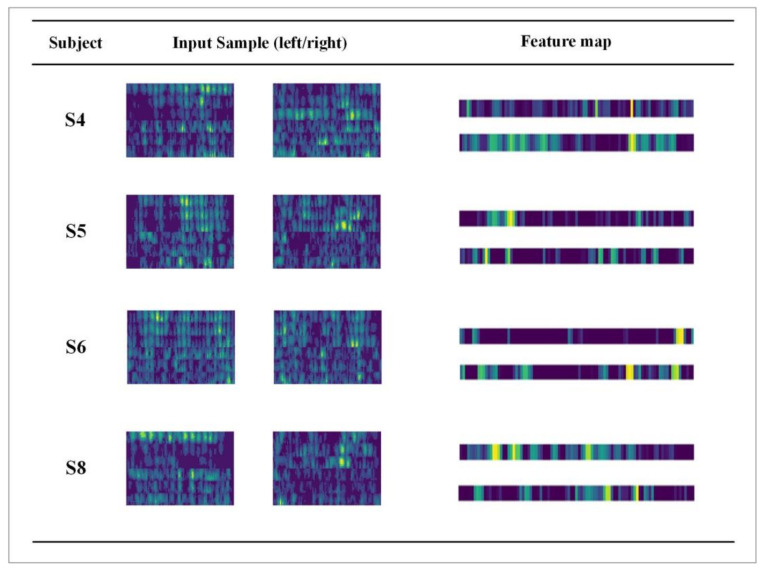
Visualization of the feature maps of the randomly selected subjects.

**Figure 9 entropy-24-00376-f009:**
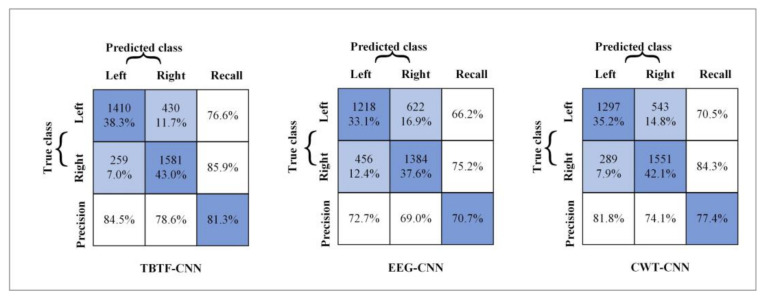
Confusion matrix for different network architectures.

**Figure 10 entropy-24-00376-f010:**
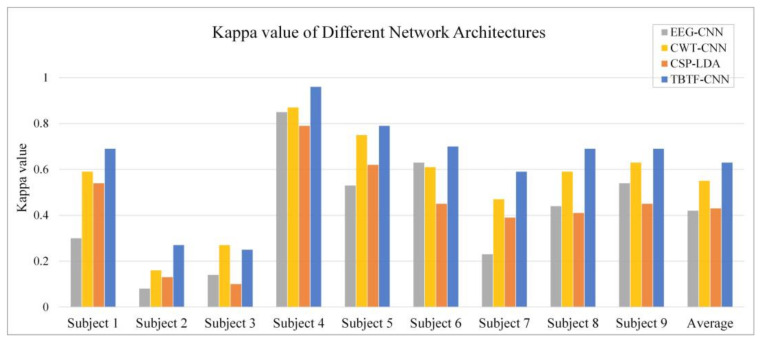
The kappa value of different network architectures.

**Figure 11 entropy-24-00376-f011:**
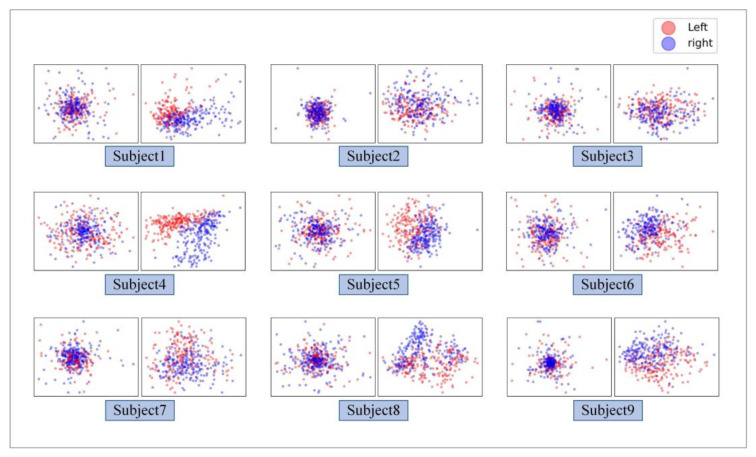
The feature distribution of all subjects before and after.

**Table 1 entropy-24-00376-t001:** Implementation details for proposed TBTF-CNN architecture.

Layer Type (EEG/CWT)	Number of Filters	Size of Feature Map	Kernel Size	Stride	Parameters
Input layer					
		1 × 875 × 6			
		64 × 93 × 1			
Convolutional layer					
	32	1 × 110 × 32	1 × 8	1 × 8	1568
	32	1 × 93 × 32	64 × 1	1 × 1	2080
Max pooling layer					
	32	1 × 36 × 32	1 × 3	1 × 3	
	32	1 × 31 × 32	1 × 3	1 × 3	
Flatten layer					
		1152			
		992			
Concatenate layer		2144			
Fully connected layer (Dropout = 0.8)		128			274,560
Output layer		2			258

**Table 2 entropy-24-00376-t002:** Comparison table of the proposed method with other methods.

	Ang et al. [21]	Tabar et al. [29]	Huang et al. [41]	Zheng et al. [42]	Raza et al. [43]	Our Method
Dataset	2b	2b	2b	2b	2b	2b
S1	70.0	76.0	74.3	72.5	70.3	**84.5**
S2	60.0	**65.8**	61.8	56.4	50.6	63.3
S3	61.0	**75.3**	66.5	55.6	62.8	62.3
S4	97.5	95.3	91.5	97.2	93.8	**98.1**
S5	**92.8**	83.0	79.5	88.4	63.8	89.7
S6	81.0	79.5	74.8	78.7	74.1	**85.0**
S7	77.5	74.5	71.4	77.5	61.9	**79.5**
S8	**92.5**	75.3	73.3	91.9	83.1	84.7
S9	**87.2**	73.3	72.0	83.4	77.2	84.5
AVG	80.0	77.6	73.9	78.0	69.7	**81.3**

## Data Availability

The EEG data used to validate the experimental results can be obtained from http://www.bbci.de/competition/iv/#download.

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
