# Peer review of "A Two-Branch CNN Fusing Temporal and Frequency Features for Motor Imagery EEG Decoding"

_entropy, 2022, doi:10.3390/e24030376_

Round 1

Reviewer 1 Report

The paper reports about the development of a two-branch convolutional neural network to classify MI-EEG signals, exploiting the CWT of the EEG signals. The paper is interesting, innovative and well written. In my opinion only few minors should be addressed before publications.

  • In my opinion, it should be better to move some parts from the Results section to the discussion. For instance, form line 315, there is a comparison with the procedure of Tabar. if the Authors agree, maybe, it could be moved in the discussion section.
  • In the Discussion section, it should be stressed the possibility to employ this method in BCI applications, as stressed in the abstract. The evaluation of the CWT in a limited temporal window could be a problem for this kind of application? Please, discuss this possibility in the Discussion section.
  • It could be interesting to report a graph of the accuracy of the CNN as a function of the epochs. Moreover, it could be worth to investigate a statistical difference between the accuracy of the TBTF-CNN with respect to CWT-CNN and EEG-CNN.

Author Response

Many thanks for your valuable comments and suggestions. It is my great honor receiving your recommendation. According to your suggestion, the article has been modified as follows:

1) In line 315, we move some parts compared to the Tabar method to the discussion section (line 436);

2) In the discussion section, we analyze the influence of STFT and CWT in a limited temporal window (line 448);

3) In line 467, we add the future work section. In the following study, we will use data augmentation techniques to improve the quality and quantity of EEG data to investigate the statistical differences in the accuracy of different models.

Reviewer 2 Report

entropy-1611317: “A Two-branch CNN Fusing Temporal and Frequency Features for Motor Imagery EEG Decoding”

In this clearly written manuscript, the authors develop the powerful EEG approach improving the motor imagery-EEG decoding accuracy. All essential mathematical modifications are described successively, cogently and intelligibly. Conclusions are supported with obtained data. This manuscript could be recommended for publication even in the current form.

Remarks/recommendations:

1) abbreviations of “MI” and “SNR” (lines 10 and 65, respectively) need their full forms;

2) a separate list of all abbreviations would be very useful;

2) the reference number for “Tabar et al.” is omitted (line 430);

3) the legend to Figure 11 seems to be truncated (line 385)

Author Response

Many thanks for your positive comments and suggestions. It is my great honor receiving your recommendation. According to your suggestion, the article has been modified as follows:

1) We complete the abbreviations "MI" and "SNR" (line 10 and 65, respectively);

2) At the end of the article, we add a list of abbreviations (line 484);

3) We complete the reference number of "Tabar et al." (line 435);

4) We add the legend of Figure 11, with red representing the left hand and blue representing the right.